# Using Behavioural Insights to Improve the Uptake of Services for Drug and Alcohol Misuse

**DOI:** 10.3390/ijerph18136923

**Published:** 2021-06-28

**Authors:** Hayley Alderson, Liam Spencer, Stephanie Scott, Eileen Kaner, Alison Reeves, Sharon Robson, Jonathan Ling

**Affiliations:** 1Population Health Sciences Institute, Newcastle University, Newcastle NE2 4AX, UK; hayley.alderson@newcastle.ac.uk (H.A.); Steph.scott@newcastle.ac.uk (S.S.); Eileen.kaner@newcastle.ac.uk (E.K.); 2Hartlepool Borough Council, Civic Centre, Hartlepool TS24 8AY, UK; Alison.Reeves@hartlepool.gov.uk (A.R.); Sharon.Robson@hartlepool.gov.uk (S.R.); 3Faculty of Health Sciences and Wellbeing, University of Sunderland, Sunderland SR1 3SD, UK; Jonathan.ling@sunderland.ac.uk

**Keywords:** drug and alcohol treatment, qualitative, quantitative, EAST framework, behavioural insights

## Abstract

In the U.K., 270,705 adults were in contact with drug and alcohol treatment services between April 2019 and March 2020. Within the same time period, 118,995 individuals exited the treatment system, and just over a third (36%) left treatment without completing it. The latter includes individuals declining further treatment and unsuccessful transfers between services. The aim of this study was to explore the factors that affect drug and alcohol treatment uptake within a drug and alcohol service in North East England. A mixed-methods approach was adopted. The exploration of factors affecting treatment uptake was captured through a behavioural insights survey and 1:1 in-depth qualitative interviews with service users within one council area within the North East of England. There were 53 survey participants, and a further 15 participants took part in qualitative interviews. We triangulated data sources to report consistencies and discrepancies in the data. Findings show that treatment services aiming to reduce missed appointments and increase retention rates need to implement several strategies. Consistently distributing appointment cards, using text message reminders, displaying a timetable presenting all treatment options, and displaying information in a format to ensure it is accessible to individuals with lower health literacy and reducing wait times for appointments will all improve appointment attendance.

## 1. Introduction

Around 10.4 million U.K. adults consume alcohol at levels that increase their potential for health-related harms, and 2.7 million adults have taken an illicit drug within the last year [1]. In England and Wales in 2019, there were 7565 deaths registered that related to alcohol-specific causes (alcohol-specific deaths include health conditions where each death is a direct consequence of alcohol misuse) [2] and 4393 registered deaths relating to drugs [3,4]. The North East has the highest death rate across the English regions, at 16.6 alcohol deaths per 100,000 compared to 7.9 per 100,000 in London. Additionally, the North East has the highest rates of deaths related to drug misuse at 134.2 males and 57.1 female deaths per million compared to the East of England, which has the lowest rates at 49.0 males and 18.4 female deaths per million [4]. This highlights the significant level of need for agencies such as drug and alcohol services in the North East. The annual cost of alcohol-related harm is approximately £21.5 billion, and illicit drug misuse in the U.K. is approximately £10.7 billion [5,6]. These costs span health and social care, crime, and loss of productivity in the workforce, therefore specialist services are required to contribute to the prevention and treatment of substance-related harm and keep the burden of care away from other parts of the health and social care system.

In the U.K., there were 270,705 adults in contact with drug and alcohol treatment services between April 2019 and March 2020 [7]. Of the population in receipt of treatment, 98% of individuals received a psychosocial intervention, and 56% received at least one pharmacological intervention [7]. Within the same time period, 117,678 individuals exited the treatment system, with just under half (47%) discharged having successfully completed their treatment [7]. Additionally, 36% of individuals left treatment without completing it (this includes individuals declining further treatment and unsuccessful transfers between services). At the end of March 2019, over a third (40%) of individuals still in treatment reported four or more treatment episodes [7]. The initial approach and early interactions with service users play key roles in uptake and retention within drug and alcohol services, with around 50% of clients failing to attend their second treatment session [8,9]. For treatment programmes to be effective, they typically need to incorporate multiple different components [10].

When considering treatment adherence, a large body of literature has examined adherence to maintaining a medical regime for chronic conditions such as asthma, diabetes, HIV/AIDS [11,12,13,14]; however, less is known about adherence to substance use treatment, inclusive of both substitute prescribing and psychosocial interventions. The WHO definition of treatment adherence is “the extent to which a person’s behaviour, taking medication, following a diet, and/or executing lifestyle changes- corresponds with the agreed recommendations from a healthcare provider” [14]. An individual’s ability to adhere to a treatment plan can be compromised by multiple factors such as the absence of social support, socio-economic status, provider-patient communication and relationships [15,16], stigma [17,18], and comorbidities [12,19]. A systematic review conducted by Brorson et al. (2013) highlighted that out of 122 studies, only 4% considered the risk factors associated with the treatment programme, such as the treatment setting, and only 5% considered predictors of dropout such as motivation, treatment satisfaction, and alliance [20]. When individuals do not engage fully or fail to attend treatment sessions, it leads to poorer health outcomes, risk of relapse, and a decreased likelihood of achieving recovery [11]. To achieve better long term outcomes, an individual needs to receive medium to long term support to maintain abstinence [21]. A relapse can have a high cost for the individual, their family, and the community [22]. Therefore, understanding the factors that promote or inhibit drug and alcohol treatment uptake and attendance at social and psychological interventions is key.

In this small-scale exploratory project, we adopted a behavioural insights approach to consider this problem, as this approach can contribute to improving policies and systems [23]. Behavioural insights have been used within public services and encourage individuals to make healthier choices for themselves. The behavioural insights team developed the EAST framework [23], which suggests that if you want to encourage individuals to undertake a behaviour, it should be easy, attractive, social, and timely (EAST) [24]. The current study occurred within a drug and alcohol service in the North East of England. This service offers support via neurological, biological, psychological, and sociological (NBPS) interventions and clinical interventions inclusive of substitute prescribing where appropriate. The NBPS interventions include opportunities to be involved in, for example, mindfulness, food and mood, acupuncture, SMART (self-management and recovery thinking), PAWS (post-acute withdrawal symptoms) and mutual aid groups and uses cognitive behavioural therapy, motivational interviewing, and counselling approaches amongst others. The aim of this study was to explore the factors that promote or inhibit drug and alcohol treatment uptake and adherence within the drug and alcohol service and use the findings to generate easy-to-implement interventions to improve service outcomes and contribute to reducing the alcohol and drug-related harm individuals experience.

## 2. Materials and Methods

### 2.1. Design

A concurrent mixed-methods approach was adopted. The exploration of factors affecting treatment uptake was captured through a behavioural insights survey collected between September 2018 and January 2019, which is reported here as Phase 1, and qualitative interviews with service users conducted between October 2018 and May 2019 reported as Phase 2 within one council area within the North East of England. Data sources were triangulated to ensure that consistencies and discrepancies in the data were reported.

#### 2.1.1. Phase 1: Behavioural Insights Survey

A behavioural insights framework, EAST [23], was used to assess how easy, attractive, social, and timely the treatment offered was perceived to be by adult service users. Table 1 shows how each element was conceptualised. Within this framework, a survey was conducted with individuals accessing the same substance use service in the North East to explore their views and experiences of the current service and identify areas where improvements may be made in the future. Potential factors that could influence uptake and continuation of substance use treatment were identified by the research team and drug and alcohol practitioners involved in the study and mapped on to the EAST framework.

The survey was made available both online and in paper-based copies and was completed by service users accessing the drug and alcohol service, although all completed surveys were on the paper version. The survey comprised seven questions on demographic information: age, sex, employment status, marital status, and number of dependents alongside information regarding an individual’s substance use and treatment status. A further 43 statements focused on perspectives of treatment inclusive of NBPS and clinical interventions. These were completed using a 7-point Likert scale, with responses ranging from 1 “strongly disagree” to 7 “strongly agree” (see Appendix A). The survey also included a section for free text responses to provide participants with an opportunity to provide any further comments about the service that they felt were important.

#### 2.1.2. Phase 2: Qualitative Interviews

Service users were interviewed to explore their experiences of the drug and alcohol service provision. Semi-structured interviews were structured around the EAST framework. We explored how easy it was to access the service (referrals in, methods of communication); attractiveness of the service (participants emotional response to the service and what they liked/disliked about it); social aspects of the service (perceptions of others receiving treatment and treatment centre staff) and timely (participants motivation to attend/or not and what influenced their decisions). All interviews were carried out by the same researcher (LS). The interviews occurred at a time and location convenient to the service user and in a private room to ensure confidentiality. During interviews, participants’ views were reflected back to them to ensure our understanding and interpretations were correct. Interviews were audio-recorded and transcribed verbatim. Once interviews were complete, all participants were debriefed. All participants had the opportunity to ask questions and were assured that they could speak to their allocated keyworker if they required further support. In addition, participants were provided with a list of additional sources of help and support, should they wish to seek it.

### 2.2. Sampling

The study occurred within an established drug and alcohol service with two treatment centres within the North East of England. The practitioners working within the service and an embedded researcher (LS) recruited service users. The researcher was embedded within the two centres, one centre focused on clinical treatment (e.g., substitute prescribing) and the other focused on community treatment (e.g., delivery of therapeutic NBPS interventions and group work). The researcher attended the services on different days and times to recruit current and recent service users. Sampling in this way meant that practitioners and the researcher could ask all service users as they accessed the service if they would like to participate, which included a spectrum of service users ranging from those attending for an initial assessment through to those in receipt of a maintenance prescription.

#### 2.2.1. Phase 1: Behavioural Insights Survey Study Participants

The age range of participants was 22–64 years (mean 40.5 years). Two-thirds of respondents were male (n = 35), and a third were female (n = 18). NDTMS data for the service showed that it was approximately 2/3 males and 1/3 females accessing the service with a mean age of 38.5 years, so the demographics of recruited participants corresponded with the demographics recorded for the local NDTMS data set.

Most participants were single (n = 43; 81.1%), 3 reported being married, 3 co-habiting, 3 were in a relationship, and one did not disclose their relationship status. The drugs that participants reported using included alcohol, cannabis, opioids, cocaine, amphetamines, hallucinogens, and benzodiazepines. Most participants were unemployed (n = 42; 79%), five reported working, and two were in part-time education. Most participants were currently receiving treatment (n = 43; 81%), one participant had completed treatment, and five were in the process of initiating treatment/completing an initial assessment.

#### 2.2.2. Phase 2: Qualitative Interviews Study Participants

All participants were 18 years and over; 11 participants were male and four female. Participants reported using heroin, alcohol, cannabis, cocaine, and amphetamines, as well as misusing prescription drugs. They also spoke about drug treatment and substituted prescribing in the forms of methadone and Subutex. All participants were white British, in keeping with the local demographic. Interviews lasted 10–50 min (mean: 30 min).

### 2.3. Data Analysis

#### 2.3.1. Phase 1: Behavioural Insight Survey Data

A measure of central tendency (mean) and variability (SD) were calculated for each factor. Forty-three statements (using a 7-point Likert scale) organised around the EAST framework were completed. These provide an overall gauge of responses. Means over the midpoint of 4 (neither agree or disagree) indicated agreement or positive responses. However, measures of central tendency are not a suitable representation of data when responses are polarised (e.g., if some service users strongly agree and some strongly disagree with a statement). For this reason, the number and percentage of responses at the extremes of each scale; strongly disagree/disagree (scores ≤ 2) and strongly agree/agree (scores ≥ 6) were also calculated. In order to provide some gauge of consistency in responding [25], two of the survey questions appeared twice. Responses to repeated questions did not significantly differ. Anonymised data were analysed using SPSS, and summary statistics were calculated.

#### 2.3.2. Phase 2: Qualitative Interviews

Anonymous data were analysed in NVivo 11. The analysis of qualitative interview data was based on the EAST framework [23] as per the quantitative analysis above. Factors that could influence the uptake and continuation of substance misuse treatment were identified and mapped onto the framework. Framework analysis [26] based on the EAST framework provided a structure that helped the research team to summarise the data in a way that concisely answered the research question [27]. A spreadsheet was used to generate a matrix using the EAST framework components, and data were “charted” into the matrix. Two researchers (HA and LS) conducted the qualitative analysis and populated the framework, including references to illustrative quotes prior to circulating to the full team for their consideration. Direct quotes presented within the report came from service users. Participant identifiers have been used throughout to protect participant identities.

### 2.4. Consent Process and Ethics

Prior to each survey or interview, participants were provided with an information sheet detailing the purpose of the study and their rights as participants, inclusive of their right to withdraw, the voluntary nature of participation within the study, and their right to anonymity. Informed consent was obtained from all participants. This was completed by ticking a box prior to completing the survey or interview. Interviews were audio-recorded with consent and transcribed verbatim. Transcripts were anonymised, and an individual identifier was allocated to each transcript. Each participant received a £5 voucher for completing the survey and a £10 voucher for taking part in the interview.

Ethical approval was gained from the Newcastle University Faculty of Medical Sciences and Hartlepool Borough (Ref: 1528_1/2018).

## 3. Results

### 3.1. Phase 1: Survey

#### 3.1.1. Sample Size/Response Rate

A survey was conducted with 53 service users (approximately 25% of the available cohort at the time of the study).

#### 3.1.2. Ease of Attending Appointments

Nearly two-thirds of participants (63.5%) agreed/strongly agreed that it was easy for them to attend appointments, while 31.3% of participants agreed/strongly agreed that attending appointments took a lot of energy and effort. A fifth (22.4%) of participants agreed/strongly agreed that attending appointments could be expensive. Most participants (72%) agreed/strongly agreed that receiving a text message reminder the day before an appointment would make it easier to remember, and half of the participants (50%) also agreed that it would be useful to receive appointment letters in the post.

#### 3.1.3. Attractiveness of Treatment

The majority of participants responded to questions regarding the service positively. Participants reported agreeing/strongly agreeing that; they were made to feel welcome when they arrived (80.4%), staff were easy to talk to (78.8%), staff explained things well (78.9%), and they felt that their privacy would be protected (77%). However, participants also agreed/strongly agreed that they would like further information about what would happen once they accessed treatment (40%), and 25.5% of participants agreed/strongly agreed that they felt on edge when accessing the service.

#### 3.1.4. Timeliness of Treatment

A third (36.6%) of participants agreed/strongly agreed that they had to wait a long time for their first appointment, and a fifth (19.3%) that they had to wait a long time between appointments. Although the treatment was widely seen as beneficial, almost half (48%) agreed/strongly agreed that they would be more likely to attend an appointment if they knew how much it costs to provide.

Respondents rated their motivation to change very highly. Participants agreed/strongly agreed that they perceived treatment would be beneficial to them (90.2%), 85.4% agreed/strongly agreed that they were motivated to change, confident that they could change (74%), and felt prepared to take up treatment (88.3%).

#### 3.1.5. Social Aspects of Treatment

Over half of respondents agreed/strongly agreed that friends were aware of (64%) and supportive of (61.7%) them accessing treatment, and three-quarters agreed/strongly agreed that their family was aware of (85.5%) and supportive of (74.5%) of them accessing treatment. However, approximately a third of respondents agreed/strongly agreed that substance use was a part of their identity (34%) and over half agreed/strongly agreed that it was a big part of their life (52.9%) [25]. Respondents agreed/strongly agreed that they have commonalities with other service users (63.5%). Table 2 shows the responses to the survey items.

### 3.2. Phase 2: Qualitative Interviews

#### 3.2.1. Ease of Attending Appointments

Interviews were conducted with 15 service users. Within the qualitative interviews, participants described their desire for services to be more flexible in their approach to offering appointments and accommodating the needs of service users with families or work commitments. Additional opening hours in evenings and/or weekends were described as helpful.


*People with different situations, kids at home, trying to balance maybe a job, they don’t want to lose their mortgage. Weekends would be massive, to open at a weekend, you know. Especially those people that are juggling work life and home life*
(Service user 9, male)

Several service users felt that reminders for appointments would be useful, especially if there had been a long gap since their previous appointment. Appointment cards that were handed out were often mislaid or forgotten about.


*No reminders or anything like that. You know, if they give you an appointment card for a month’s time, how the fuck are you going to remember that?*
(Service user 12, male)

Participants did, however, offer solutions to this, such as automated text reminder services like those used for GP and dentist appointments.


*Maybe if they had an automated text system that just sent you a text the day before, and then you could ring up if you couldn’t make it and change your appointment or just reply, “Yes,” to it, or whatever.*
(Service user 9, male)

Participants raised concerns regarding communication between the service provider and service users. One participant described an occasion when they had paid for a taxi to attend an appointment, and when they arrived, there was a poster on the door stating that the session had been cancelled.


*If people are coming from over the other side of town, in bad weather, [only] to find that there’s just a note on the door to say that the meeting is not going ahead, it’s not very good, is it?*
(Service user 6, female)

#### 3.2.2. Attractiveness of Treatment

There were several positive factors attributed to the available service. The Food and Mood classes were viewed positively. These classes provided individuals recovering from drug and alcohol misuse with an insight into how food can affect mental health and addiction and what a balanced nutritional diet looks like. The classes were co-facilitated between service users and staff members.

*I do the Food and Mood classes on a Friday with [staff member], so I teach the guys how to cook a little bit. It’s just nice, isn’t it? Giving something back is like therapy*. (Service user 9, male)

Despite generally positive views, some participants who had been through the programme (or similar ones) described feeling as though they were not learning anything new.


*Because of my knowledge, and the amount of time I’ve been in services, I do tend to find them quite tedious and boring in all honesty. It’s nothing to do with the staff or the content, it’s because I know it all and it’s a bit like sucking eggs.*
(Service user 11, male)

When considering attractiveness, service users discussed the service sites and whether they perceived them to be secure, safe, and well situated. The service was offered from two different locations, which evoked very different responses from participants. One site was disliked by most participants, the main reason being they felt stigmatised attending a place aimed solely at drug rehabilitation (rather than, say, a health centre).


*I didn’t like going there anyway because of how it looked outside when people are under the influence [of drugs] outside. I didn’t want to be tarred with that brush.*
(Service user 6, female)

The same site also raised fears regarding personal safety, with two respondents reporting having been attacked at this location, and another reported being so scared of the place that they declined treatment when it was offered.


*I went there [one of the rehabilitation sites] a long time ago before I came here. And I was speaking to a lovely lady called [staff member]. But then I stopped going because there was somebody outside, and they’d obviously been on something, and I got attacked outside. So, I didn’t go back.*
(Service user 6, female)

#### 3.2.3. Social Aspects of Treatment

When considering support via informal support networks such as other service users, participants described feeling as though group sessions were beneficial as they enabled individuals to share experiences and receive peer support.


*But that’s what the group is good for, because I shared what happened to me in the groups and that does help people. It’s seeing different people’s perspectives and different people’s way of dealing with things. You can relate to that and gain a lot from it.*
(Service user 9, male)

The social aspect of recovery was perceived as important for participants. The concept of talking to other people experiencing similar problems was deemed as positive. Peers within treatment were identified as a positive source of friendship and a good support network.


*Then, my neighbor next door, [name], he comes here as well. He’s been clean the same [length of] time. We’ve, sort of, helped each other. He’s been clean the same time as me […] He copes, I cope, we help each other. I’ve found that a bit easier this time as well, instead of being by myself.*
(Service user 14, male)

Participants also felt that services needed to accommodate service users according to where individuals were in their recovery journey to reduce the exposure to individuals who are still actively using, which may jeopardise their progress.


*You have people who don’t need that trigger. They don’t need to come into a building and see people who are a little bit further back in their recovery, who still possess the same sorts of traits, the look, the way they carry themselves, they still talk constantly about their substance, about where they can get it from. Then we’ve had it in here, there’re people still outside or inside sorting out [drug] dealing and stuff like that. People who are struggling with their recovery, but are serious about it, don’t want to be with people who are in the building and not really serious about their recovery.*
(Service user 4, female)

#### 3.2.4. Timeliness of Treatment

Although participants felt that the time spent waiting for an appointment was longer than it ideally would have been, there was a recognition that services were stretched.

*Obviously, they’ve got quite a hefty caseload, and appointments are not readily available*. (Service user 11, male)

There was a recognition from participants that once they had a keyworker allocated, workers did try to be responsive to service users, even if it was via a text message or a phone call.

*He’s usually quite quick when it comes to responding [to texts or calls]; it’s just actually trying to get an appointment [that’s the problem]*. (Service user 3, male)

Some participants did report being seen quickly; however, they also stated that this occurred as a result of a cancellation rather than as standard practice.


*They actually had a cancellation or something, so they were able to see me more or less straight away, which was helpful.*
(Service user 9, male)

Closely linked to the timeliness of the treatment offered was the motivation and readiness to change on behalf of the service user. Often participants stated that they entered treatment as a direct consequence of worsening physical and/or mental health or a breakdown in family relationships. One participant stated that *“It’s life or death”* (Service user 10, male). Others stated that it was because they were *“sick of being the way they were”* (Service user 5, male).

## 4. Discussion

Although the service in this study had one of the best treatment outcomes for service users within the region, it only had a 56.5% uptake rate for those assessed and offered support. Therefore, this study explored the factors that promote or inhibit engagement with the service using the EAST framework [23], and the discussion section is organised around the framework. The findings from the study were incorporated into a service redesign to help increase the uptake of NBPS and clinical treatment interventions.

### 4.1. Ease

There was a discrepancy when considering how easy it was for participants to attend appointments. Where the survey results showed that two-thirds of respondents agreed/strongly agreed that it was easy for them to attend appointments. The interviews highlighted that many participants would like more flexibility in appointments, inclusive of evening and weekend openings to accommodate the varying needs of service users. There was more consistency between the survey and qualitative findings in that participants felt that receiving text message reminders and/or letters in the post would be beneficial in making it easier to remember appointments. The use of SMS (text message) reminders within healthcare settings have demonstrated promise [28,29,30], and a systematic review [31] found that using SMS reminders increased the likelihood of attendance at clinical appointments by 50%, compared to no appointment reminder. Mobile (cell) phones are widely available among service users accessing drug and alcohol treatment services. One study found that in a sample of 389 drug and alcohol service users, 83% owned a mobile phone [32]. SMS reminders are therefore a viable method of communication, and they offer a simple and cost-effective option to improve service attendance and, in turn, service delivery [31].

Non-attendance rates are generally high within substance misuse services [33]. Service users who regularly miss appointments have poorer treatment outcomes, ultimately resulting in being discharged/dropping out of treatment [32]. This is problematic not just because of the effect on rehabilitation but also from an economic perspective as it results in inefficient use of resources.

Accessibility of appointments was also identified as problematic for some of the service users within this study for logistical reasons. These included a reliance on public transport, childcare needs, and costs associated with attending, such as transportation costs. These findings corroborate the findings of previous research [34]. Clients participating in this study had to attend a service site within the town centre, which may have meant that they had to travel some distance to access treatment. Previous research has shown that when clients live in an area that is more geographically dispersed, they have fewer public transport options [35], which can affect attendance. Individuals with shorter travel distances have improved treatment completion rates [36]. A solution to this would be for services to think creatively regarding appointments, potentially working on an outreach basis, utilising community centres and general practitioner surgeries, for example, to minimise the distance service users must travel to attend appointments. In addition, given the COVID-19 pandemic that has occurred since the completion of the study, the potential of offering services online via a digital method could be considered [37]. However, it must also be acknowledged that a significant proportion of the population groups accessing drug and alcohol services may experience digital poverty and may be less likely to have access to a smartphone, may be on a pay as you go contract, or have a data plan that makes it more expensive to access the Internet [38].

### 4.2. Attractiveness

The location and environment within the treatment agencies were identified as important and influential when accessing treatment. The concept of feeling “unsafe” was consistently reported in both the survey and in qualitative interviews, with a subset of participants indicating that the treatment services made them feel “on edge” or “threatened”. Treatment agencies aiming to improve attendance rates should consider service delivery locations and explore how to support individuals to feel more relaxed when accessing support, which in turn would influence appointment attendance. Such support may come at a financial cost, such as upgrading or making alterations to existing facilities, but this could become more economical if it led to greater engagement with services.

The data regarding the “attractiveness” of the service offered were contradictory in places. Individuals reported being made to feel welcome, staff were easy to talk to, and things were described well. However, approximately half the survey participants stated that they would have liked to have received more information regarding the services available and what to expect from treatment. This finding raises an important question regarding the health literacy of service users who were accessing the drug and alcohol service as information leaflets were available, which provided this information. Individuals with low education achievement [39] and weak social connections [40] are more likely to have limited literacy levels. Public Health England reported that in England, up to 61% of the working-age population find it difficult to understand health and well-being information [41]. Limited health literacy may directly affect an individual’s ability to be involved in decision-making regarding their health. Therefore, potentially vulnerable and disadvantaged groups such as those accessing drug and alcohol services are likely to be adversely affected. Future initiatives should seek to understand the literacy levels of individuals accessing treatment and then consider the best ways to communicate information to people contemplating or accessing services. Participants with multiple episodes of treatment also stated that they felt that sessions were repetitive. This is an important issue to consider as statistics show that of those individuals accessing treatment in 2019–2020, 40% had experienced four or more treatment episodes, and 26% had been involved in treatment continuously [7]. Many service users experience relapse despite multiple treatment episodes, and it is reasonable to foresee that they may need increasingly novel approaches to successfully engage them in treatment [42].

### 4.3. Social

It is recognised that individuals problematically using drugs and/or alcohol are often subject to stigmatisation. This can paradoxically lead to continued use once an individual has entered the substance-using culture, as they are accepted by peers in a similar situation. Participants within the current study found that peer support and the process of giving and receiving advice and support from individuals in a similar circumstance were mutually beneficial. This finding has been established in previous work. Peer and social network support are a key component of many drug and alcohol treatment and recovery approaches [43,44,45]. Active engagement with mutual aid and/or peer support groups plays an important role in recovery [46,47]. The opportunity to involve supportive social networks whose attitudes and behaviours are congruent with an individual’s recovery goals in treatment also helps to mobilise an individual’s social recovery capital [48,49]. Social recovery capital is the amount of internal and external resources that an individual can draw upon to initiate and sustain recovery from problematic drug and alcohol use [50,51].

We did find tension regarding the exposure to individuals at different stages of recovery. Examples included individuals whose primary substance of choice was alcohol described being exposed to individuals intoxicated with other substances such as heroin; additionally, individuals who were currently abstaining described being “tempted” to use again due to exposure to current users while accessing treatment. This is perhaps unsurprising as treatment services are accessed by individuals who are abstaining and active users and so potentially could provide ready access to substances that could facilitate continued use or relapse. Heslin et al. (2013), among others, acknowledged that cues within the environment could impact negatively on those accessing services for substance misuse, potentially triggering cravings and increasing the desire to use [52,53]. In particular, stigma may also have an influence on whether people seek treatment. While this was not the focus of the current study, this should be explored in future work. Creating the balance between the temptation to use and eliciting support from peers with lived experience is hard to manage. This is especially important as substance misuse is not something that happens to someone in isolation but takes place within a person’s world that affects and is affected by other people such as friends, family, and practitioners. An individual misusing alcohol and/or drugs has an increased chance of successfully reaching their goals if they have support from others [54]. Therefore, the social world around an individual has enormous potential to help and support them to deal with their problems. The findings from this study support previous literature showing that social support improves self-efficacy [55].

### 4.4. Timeliness

Many participants in both the survey and interviews felt the wait time for an initial appointment and/or the time between appointments was too long. It is important for the service to consider the “window of opportunity” and respond pro-actively. Previous research has shown that once an individual contacts a treatment agency and indicates that they are motivated and ready to change, the individual needs to be supported as quickly as possible. If not, the effect is detrimental [56]. Previous work has found that shorter wait times lead to fewer missed appointments [57] and that longer wait times are often a barrier to engaging in treatment and result in a lower likelihood of treatment entry [56,58].

Thus, when an individual displays motivation and readiness to change, it is important that services can respond efficiently to offer support. Prochaska and DiClemente [59] designed a stages of change model regarding substance misuse and motivation to change. The stages of this model include pre-contemplation, contemplation, determination, action, and maintenance. Often individuals consider accessing treatment if they are in the contemplation, determination, or action stages of the change model. Study participants often decided to enter treatment as a result of significant worsening of physical and/or mental health issues or a breakdown in relationships with family members.

### 4.5. Strengths and Limitations of the Study

This study used both quantitative and qualitative methods. This provided an opportunity to collect data that provided some additional depth providing a richer insight into the contextual factors that surrounded the participants, which would have been missed if we had only collected survey data. Collecting data using both quantitative and qualitative methods enabled us to triangulate data sources and highlighted issues that were reported consistently, as well as identifying discrepancies in the data sources. This study has shown that the EAST framework can successfully be applied within a drug and alcohol context. It would be beneficial to use this framework with a larger sample and to use purposive sampling to enable maximum variation to be obtained.

A limitation of the study is that the sample size for the quantitative element was small, and so we were unable to conduct a sub-group or inferential analysis. A further limitation is that the results should be interpreted with caution regarding respondents’ perspectives related to being highly motivated to attend treatment as 81% of participants were actively engaged in treatment and therefore may be more likely to respond positively. This finding will not be applicable to all service users attending drug and alcohol services. The population involved in this study were all currently engaging successfully in treatment and were therefore already showing motivation and commitment to change/address their behaviour. This does not reflect the opinions of service users who have declined to engage in available support or indeed service users who have failed to attend appointments resulting in them being discharged from services, who are likely to have very different views of treatment services. Additionally, there is potential that participants responded to the survey in a socially desirable way. While we opted to use qualitative interviews and a survey in our work, we are aware that we could have used other methods such as focus groups, observational research, a user journey map, or a quasi-experimental design, which have also been used to investigate the topic of drug and alcohol treatment.

### 4.6. Implications for Practice

Due to known challenges regarding treatment adherence, the following recommendations for practice have been proposed.

#### 4.6.1. Appointments Cards and Reminders

When arranging appointments, we recommend systematically providing an appointment card for every appointment. Where service users have a fixed address, appointment letters should be sent and an automated text message reminder the day before the scheduled appointment. Automated texts could also be used to inform service users of cancelled sessions.

#### 4.6.2. Accessibility of Appointments

Consideration should be given to how and where appointments take place. If, for example, multiple appointments are required each week, it should be considered whether they could all take place on the same day/at the same time taking a multidisciplinary approach, or whether alternative methods of engagement could be used such as telephone or online.

#### 4.6.3. Attractiveness of Treatment

In response to individuals reporting feeling “on edge” attending a specialised treatment service, services need to pro-actively improve the feel of the building by making cosmetic changes to the interior. This would help to make facilities more appealing and improve the experience of service users.

#### 4.6.4. Information regarding the Service Offer

In response to individuals receiving accessible information regarding available services and what to expect from treatment, it is important to think about the most appropriate methods of communicating information regarding available services. It is recommended that services display the timetable/provide ALL service users with details of the whole service offer (including groups etc.) and provide a clear expectation of what treatment will entail.

## 5. Conclusions

Several strategies can help treatment services reduce missed appointments and increase retention rates of service users, which will ultimately reduce the alcohol and/or drug-related harm that they experience. Consistently distributing appointment cards, using SMS reminders, displaying a timetable presenting all treatment options, and displaying information in a format to ensure it is accessible to individuals with lower health literacy and reducing wait times for appointments will all influence appointment attendance. Services need to offer initial appointments in a timely fashion, and premises should, where possible, accommodate the needs of service users at differing stages of recovery.

## Figures and Tables

**Table 1 ijerph-18-06923-t001:** EAST framework summary of factors assessed.

**EAST Framework Component**	**Description**	**Factors Assessed in Survey**
Easy	Ease of attending the service for appointments and assessments.	Cost and time required to travel to and from appointments The amount of time and effort taken up by appointments and how these fit around other commitments Understanding of what treatment would involve Usefulness of appointment reminders
Attractive	How appealing treatment is.	Perceived benefit of treatment Perceived protection of confidentiality and privacy Emotional response to treatment centre (e.g., feeling relaxed/on edge)
Social	Positive or negative views of other service users and treatment centre staff. Perceived support for treatment involvement from friends and family.	Family/friends awareness and support Perception of others receiving treatment Perception of treatment centre staff
Timely	Indications of whether the respondent is ready to change their substance use. How long they have to wait for/between appointments.	Wait time for initial appointment Wait time between appointments. Motivation to change

**Table 2 ijerph-18-06923-t002:** Summary of responses to survey items.

Survey Item	Mean (SD)	Strongly Disagree/Disagree (%)	Strongly Agree/Agree (%)
It is easy for me to get to and from appointments at the treatment centre	5.00 (1.96)	10 (19.2%)	33 (63.5%)
It is expensive for me to get to and from appointments at the treatment centre	3.43 (2.10)	24 (49.0%)	11 (22.4%)
Appointments can/do fit in around the rest of my life	5.27 (1.54)	4 (7.8%)	31 (60.7%)
Treatment and appointments will/do take up a lot of my time	4.02 (2.04)	18 (35.3%)	16 (31.4%)
Treatment and appointments will/do take up a lot of my energy and effort	4.14 (1.88)	14 (27/5%)	16 (31.3%)
It is/would be useful to receive text message reminders the day before appointments	5.74 (1.40)	3 (6.0%)	36 (72.0%)
It is/would be useful to receive appointment letters from the service	4.81 (1.70)	8 (15.4%)	26 (50.0%)
If I knew how much each appointment cost to provide, I’d be more likely to attend	4.74 (1.96)	11 (22.0%)	24 (48.0%)
Treatment will be beneficial to me	6.31 (0.99)	1 (2.0%)	46 (90.2%)
Treatment has been beneficial for other people like me	5.94 (1.23)	1 (1.9%)	41 (78.9%)
Treatment will not work for me	2.04 (1.21)	41 (83.7%)	2 (4.1%)
Treatment doesn’t work for people like me	2.26 (1.50)	37 (74.0%)	3 (6.0%)
Staff explained things to me well	5.58 (1.87)	8 (15.4%)	41 (78.9%)
I would have liked more information about what would happen	4.24 (1.59)	11 (22.0%)	16 (40%)
I had to wait a long time for my first appointment	3.69 (2.25)	26 (50.0%)	19 (36.6%)
I have to wait a long time between appointments	3.31 (1.97)	25 (48.1%)	10 (19.3%)
I am motivated to change my substance use	6.17 (0.83)	0 (0.0%)	41 (85.4%)
I am confident I can change my substance use	6.00 (1.11)	1 (2.0%)	37 (74.0%)
I feel prepared to take up treatment	6.20 (0.96)	1 (2.0%)	45 (88.3%)
I have experienced negative effects of substance use	6.15 (1.27)	2 (3.8%)	44 (84.6%)
I want to make the most of the treatment offered to me	6.40 (0.66)	0 (0.0%)	49 (94.3%)
Now is the right time for me to change my substance use	6.25 (1.06)	1 (2.0%)	46 (90.2%)
My family are aware that I have been referred to the treatment service	6.02 (1.44)	3 (6.3%)	41 (85.5%)
My friends are aware that I have been referred to the treatment service	5.28 (1.91)	9 (18.0%)	32 (64.0%)
My family are supportive of me attending treatment for substance use	5.91 (1.41)	2 (4.3%)	35 (74.5%)
My friends are supportive of me attending treatment for substance use	5.51 (1.76)	5 (10.6%)	29 (61.7%)
I have a lot in common with others attending substance use treatment	5.40 (1.72)	7 (13.5%)	33 (63.5%)
I’m not like other people who are getting substance use treatment	3.04 (1.76)	28 (54.9%)	7 (13.7%)
I have a positive view of people seeking treatment for substance use	6.10 (1.20)	2 (3.9%)	44 (86.3%)
Substance use is part of who I am	4.20 (2.01)	15 (30.0%)	17 (34.0%)
Substance use is a big part of my life	4.75 (1.95)	12 (23.5%)	27 (52.9%)
Staff at the treatment centre are friendly	5.96 (1.43)	2 (3.8%)	41 (78.9%)
Staff at the treatment centre treat me with respect	5.98 (1.37)	2 (3.8%)	40 (77.0%)
Staff at the treatment centre are easy to talk to	6.00 (1.33)	2 (3.8%)	41 (78.8%)
Staff at the treatment centre are non-judgemental	5.54 (1.74)	5 (9.6%)	36 (69.3%)
I was made to feel welcome when I arrived	5.90 (1.50)	4 (7.8%)	41 (80.4%)
My privacy will be protected	5.79 (1.50)	3 (5.8%)	40 (77.0%)
Things I share with staff during treatment will remain confidential	5.85 (1.43)	3 (5.8%)	41 (78.8%)
I feel I can be honest with staff at the treatment centre	5.87 (1.62)	6 (11.5%)	42 (80.7%)
I have experienced negative effects of substance use	6.08 (1.40)	3 (5.8%)	41 (78.9%)
I feel “on edge” or anxious at the treatment centre	3.88 (2.08)	19 (37.3%)	13 (25.5%)
I feel relaxed and comfortable at the treatment centre	5.06 (1.79)	6 (12.0%)	24 (48.0%)

## Data Availability

The data sets used and/or analysed during the current study are available from the corresponding author on reasonable request.

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
