# Peer review of "Using Behavioural Insights to Improve the Uptake of Services for Drug and Alcohol Misuse"

_ijerph, 2021, doi:10.3390/ijerph18136923_

Round 1
Reviewer 1 Report
I found this to be a well written and interesting article on an important topic. My suggestions for improvement are as follows:
- Table 1: please ensure that information lines up across the rows so it is clear which information relates to each of the EAST Framework components
- Line 104 mentions Appendix 1, but I could not locate it in the manuscript
- Sections 2.2.1 and 2.2.2: please report the mean, standard deviation, and age range of participants in both of theses sections
- Table 2: It might help to add subheadings to separate the questions according to which EAST component they relate to; also, would be useful to include a column to show % reporting neither agree/disagree (easier that leaving it to the reader to do the maths)
- Tables 1 & 2: would be easier to read if the text in columns was left (rather center) justified
- Section 3.2: it tends to be easier to focus on quotes when they indented further than the usual text
- I think that the article would benefit if you separated out the two parts of the study, so that the Introduction section would be followed by Phase 1 (survey) Materials and Methods section and Results section, then the Phase 2 (qualitative) Materials and Methods section and Results section, followed by the Discussion section
Overall, this paper was a great read - it has informed and sparked my thinking on ways to improve AOD treatment services to ensure the needs of clients are better met. Thank you.
Author Response
Thank you for your comments, please see attached document with our responses.

Reviewer 2 Report
Thank you for the opportunity to review this paper. It is great to see work that considers the perspectives of AOD services users to improve service design and delivery.
I did find that the paper largely ignored that there is a rather large field of research on health services including appointment and treatment adherence. Although there are certainly some things that are more specific to services for AOD compared to general health services, things like stigma have been extensively covered in studies on NSPs, HIV treatment etc. When considering these broader fields, the findings are less novel, and the introduction and discussion are not well connected to existing literature. I suggest that the authors do a deeper dive into this literature and strengthen the focus on what this study adds.
In the discussion, I would also like to see more on what the implications are for practice, given that the study is fairly pragmatic in general.
Author Response

(The authors gave the same response as above.)

Reviewer 3 Report
I liked this paper but I am not sure the survey data adds very much at this point. (Small sample size and low response rate). Perhaps could form the basis of a paper presenting this as pilot or part of feasibility data.
I think the strongest element is the semi structured interviews and this should be written as a qualitative study with more detail concerning how the interview questions were derived etc. Also what steps were taken to enhance the credibility of the data. e.g was there any checks. Is there any triangulation. Could the survey data be part of the triangulation?
Author Response

(The authors gave the same response as above.)

Round 2
Reviewer 3 Report
By using the survey data to triangulate the interview data in my opinion this paper has been significantly improved and now presents some important data is a robust form.
I would just like to see a statement in the abstract and data analysis that the survey has been used to triangulate the interview data.
Author Response
Thank you for your further comments. I have added in clarification regarding triangulation as requested in the abstract (line 21) and the methods (lines 104-106).